# High-speed, three-dimensional imaging reveals chemotactic behaviour specific to human-infective *Leishmania* parasites

Rachel C Findlay[1,2], Mohamed Osman[3], Kirstin A Spence[1], Paul M Kaye[3], Pegine B Walrad[1]*, Laurence G Wilson[2]*

[1]York Biomedical Research Institute, Department of Biology, University of York, York, United Kingdom; [2]Department of Physics, University of York, York, United Kingdom; [3]York Biomedical Research Institute, Hull York Medical School, University of York, York, United Kingdom

**Abstract** Cellular motility is an ancient eukaryotic trait, ubiquitous across phyla with roles in predator avoidance, resource access, and competition. Flagellar motility is seen in various parasitic protozoans, and morphological changes in flagella during the parasite life cycle have been observed. We studied the impact of these changes on motility across life cycle stages, and how such changes might serve to facilitate human infection. We used holographic microscopy to image swimming cells of different *Leishmania mexicana* life cycle stages in three dimensions. We find that the human-infective (metacyclic promastigote) forms display 'run and tumble' behaviour in the absence of stimulus, reminiscent of bacterial motion, and that they specifically modify swimming direction and speed to target host immune cells in response to a macrophage-derived stimulus. Non-infective (procyclic promastigote) cells swim more slowly, along meandering helical paths. These findings demonstrate adaptation of swimming phenotype and chemotaxis towards human cells.

*For correspondence:
pegine.walrad@york.ac.uk (PBW);
laurence.wilson@york.ac.uk (LGW)

**Competing interests:** The authors declare that no competing interests exist.

## Introduction

Flagellar-dependent motility (*Bray, 2001*; *Beneke et al., 2019*) is key for transmission of unicellular *Leishmania* parasites, causative agents of the leishmaniases. These infections represent the world's ninth largest infectious disease burden with an expanding territory endemic to 98 countries that threatens 350 million people globally (*Alvar et al., 2012*). Like many vector-borne parasites, life cycle stage-specific differentiation is observed, affecting both *Leishmania* body shape and flagellar morphology (*Pulvertaft and Hoyle, 1960*; *Bates and Rogers, 2004*; *Ambit et al., 2011*). Notably, flagellar length changes dramatically in *Leishmania* parasites relative to other flagellated microorganisms. Procyclic promastigotes' (PCF) cell bodies are 10–12 μm long, with a flagellum of approximately the same length, while human-infective metacyclic promastigotes' (META) cell bodies are 8–10 μm in length, with a flagellum 20 μm long (*Sacks and Perkins, 1984*; *Video 1*). We summarise the key stages in the *Leishmania mexicana* life cycle in *Figure 1—figure supplement 1*.

Studying the three-dimensional swimming patterns of motile flagellates gives insight into regulation of the flagellar beat and details of the cells' navigation strategy. The physics of microorganism swimming and navigation has been considered extensively in the context of bacterial motility, where biased random walks are used to counter the randomisation of Brownian motion (*Berg and Brown, 1972*; *Taktikos et al., 2013*; *Son et al., 2016*). *Leishmania* cells are an order of magnitude larger in each dimension than typical model bacteria, and rotational diffusivity scales inversely with the cell's volume (*Berg, 1993*). In a medium with viscosity close to that of water, it takes a few seconds to randomise the orientation of an *Escherichia coli* cell, but over 200 s to randomise the orientation of

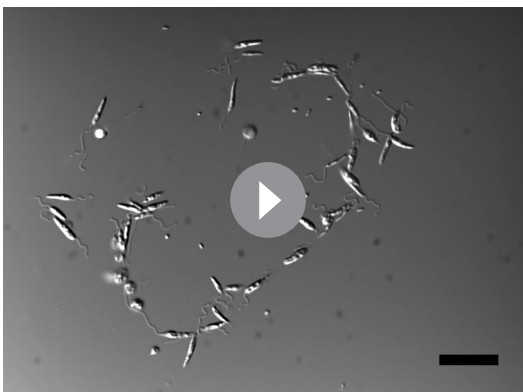

**Video 1.** Differential interference contrast movie of an unenriched sample of *Leishmania mexicana* cells, including procyclic promastigotes and metacyclic promastigotes, as well as intermediate forms. The cells were attached to the chamber surface by pre-treating the glass slide with polylysine. The scale bar represents 25 μm.

https://elifesciences.org/articles/65051#video1

*Leishmania*. The physical constraints that shape the response of these two microorganisms are therefore fundamentally different. From a fluid dynamics perspective, both operate at low Reynolds number, but Brownian motion dominates life for the bacterial system, while it is negligible for *Leishmania*. Nevertheless, there are intriguing signs that the run-tumble locomotion characteristic of bacteria like *E. coli* has analogues in motile single-cell eukaryotes: *Chlamydomonas reinhardtii* algae exhibit run and tumble behaviour (*Polin et al., 2009*), and sperm from the sea urchin *Arbacia punctulata* exhibit sharp reorientation events amidst periods of helical swimming about a straight 'run' axis (*Jikeli et al., 2015*).

Chemotaxis has been observed in trypanosomatids, although the results have often been somewhat nuanced. In an early study of chemotaxis in *L. mexicana*, *Leishmania major*, and *Leishmania donovani* (*Bray, 1983*), it was found that promastigote cells move up gradients of sugars, serum, and some serum components. This work also found some evidence that peritoneal macrophages from mice were attracted by serum that had been 'activated' by exposure to *L. mexicana*. The hypothesis that the host and motile parasites interact by diffusing messengers is supported by a study that shows that *Leishmania* promastigotes are repelled by ATP, possibly allowing them to evade neutrophils and adapt for phagocytosis (*Detke and Elsabrouty, 2008*). Other authors have found evidence that promastigotes of *Leishmania* spp. chemotax towards sugars (*Oliveira et al., 2000*), though another contemporary study suggested that the directional behaviour is more appositely ascribed to osmotaxis, and is therefore chemically non-specific (*Leslie et al., 2002*). Intriguingly, the latter study speculates that chemotactic or osmotactic stimulus could play an important role in providing an orientational stimulus for the parasites that prevents them being excreted and allows them to navigate within the sandfly host. *Barros et al., 2006* performed experiments in which they timed the duration between reorientation events in swimming cells, showing that *Leishmania amazonensis* promastigote cells respond to both osmo- and chemotaxis, and find that chemotaxis dominates the response for all but the highest stimulant concentrations. More recently, it was shown that *Leishmania braziliensis* and *L. amazonensis* show chemotactic responses towards stimulants including a repurposed cancer drug (methotrexate) (*Díaz et al., 2013*) and amino acids (*Diaz et al., 2015*). Optical tweezers studies on *L. amazonensis* (*de Thomaz et al., 2011*; *Pozzo et al., 2009*) showed that individual cells are attracted to higher glucose concentrations, and in the related trypanosomatids *Trypanosoma cruzi* and *Trypanosoma rangeli*, cells orient themselves towards tissue extracted from tsetse flies (their insect host) . Most recently, there have been exciting developments in the area of 'social motility' (collective motion) in Trypanosomes (*Oberholzer et al., 2010*; *Schuster et al., 2017*; *Shaw et al., 2019*), where it was shown that chemotactic behaviour is not just a single-cell phenomenon, but one that is exhibited in collective cell behaviour as well (*DeMarco et al., 2020*). Millimetre-scale groups of *Trypanosoma brucei* cells displaying social motility can change their swimming direction to avoid other colonies of their own species, but are attracted by chemicals diffusing from neighbouring colonies of *E. coli*.

Holographic microscopy has proven to be a versatile and powerful tool for investigating the dynamics of swimming microorganisms at high speed and diffraction-limited resolution (*Katz and Sheng, 2010*; *Romero et al., 2012*; *Weiße et al., 2020*; *Heddergott et al., 2012*; *Wilson et al., 2013*; *Molaei et al., 2014*; *Jikeli et al., 2015*; *Zhang et al., 2018*; *Thornton et al., 2020*). Three-dimensional tracking, using holographic microscopy or other techniques, unambiguously resolves chirality in the shape of objects and in the geometry of swimming paths. A long-standing question in the biological physics of the organelle at the heart of the flagellum focuses on the breaking of symmetry. The axoneme has a chiral structure in which dynein molecules on each doublet can attach to

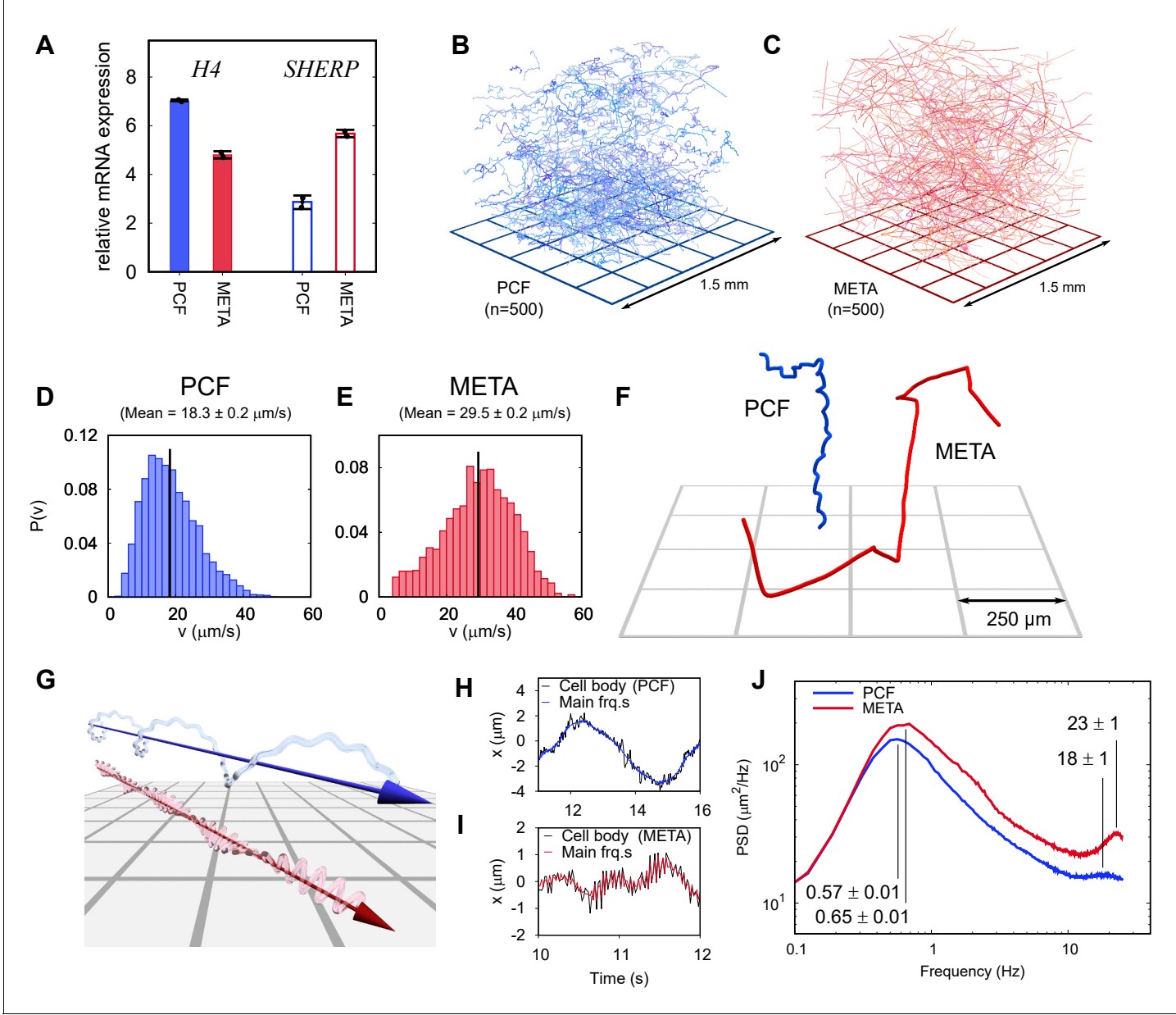

**Figure 1.** Swimming phenotypes in infective (metacyclic promastigote [META]) and non-infective (procyclic promastigote [PCF]) *Leishmania mexicana* cells. (A) qPCRs of *H4* (PCF molecular marker) and *SHERP* (META molecular marker) levels relative to *N-myristyltransferase* (*nmt*) (constant transcript control) demonstrate that enriched, but not wholly distinct populations of cells were isolated and characterised. (B) Three-dimensional swimming trajectories from 500 PCF cells of *L. mexicana*, showing stereotypical meandering helical paths (*Video 2*). (C) Trajectories of 500 META cells of *L. mexicana*. These show a marked transition to a 'run and tumble' phenotype of rapid, straight trajectories interspersed by sharp reorientations (*Video 3*). (D,E) Distribution of instantaneous swimming speeds in PCF (*n* = 3231) and META (*n* = 2202) cells, respectively. (F) Individual tracks of PCF and META cells at a smaller scale, illustrating the different swimming phenotypes (*Video 4*). (G) Cartoon illustrating the dominant swimming phenotype for PCF (top) and META (bottom). (H) The x-component of cell body motion as a function of time for a PCF cell. The black line represents the raw data, and the blue line is the motion reconstituted using the principal frequency components only. (I) The x-component of motion for a META cell, both raw data (black) and principal frequencies only (red). The flagellar beat frequency is more pronounced in META cells compared to PCF cells because the cell bodies of the former are significantly smaller, and their flagella are longer. (J) Averaged power spectra of PCF (blue, *n* = 3231) and META (red, *n* = 2202) motion, when the lowest frequencies have been removed (see text). The peaks at around 0.6 Hz correspond to the rotation of the flagellar beat plane and the peaks at around 20 Hz to the flagellar beat in both cell types. The peak values and estimated uncertainties are indicated. The online version of this article includes the following figure supplement(s) for figure 1:

**Figure supplement 1.** The *Leishmania* life cycle.

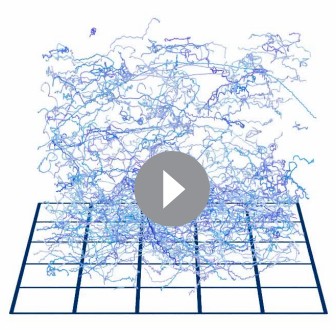

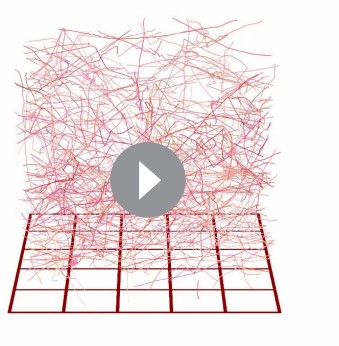

**Video 2.** Computer rendering of 500 *Leishmania mexicana* procyclic promastigote tracks, showing stereotypical meandering helical paths. The squares on the ground represent 300 µm on a side, and the tracks are between 20 and 60 s in duration.
https://elifesciences.org/articles/65051#video2

**Video 3.** Computer rendering of 500 *Leishmania mexicana* metacyclic promastigote tracks, showing the stereotypical run and tumble phenotype of unstimulated cells. The squares on the ground represent 300 µm on a side, and the tracks are between 20 and 60 s in duration.
https://elifesciences.org/articles/65051#video3

their clockwise neighbour, as viewed from the basal end of the axoneme (*Afzelius et al., 1995*). Several groups have examined whether this structural chirality of the axoneme results in chiral flagellar waveforms. *Rodríguez et al., 2009* showed that the flagellum of *T. brucei* could produce waves of either left- or right-handed chirality, in both procyclic-form and bloodstream-form cells. *Bargul et al., 2016* examined several species and strains of *Trypanosoma* and demonstrated that the chirality of the flagellar attachment can vary, even between strains of the same species, but the direction of the flagellum's rotation is independent of these morphological features. This has also been seen in the absence of an accompanying cell body attached to a flagellum; the male microgamete of *Plasmodium* shows both right- and left-handed chiralities (*Wilson et al., 2013*).

Here, we compare and quantitatively define the swimming dynamics and parameters of two key *L. mexicana* life cycle stages and find evidence of unbiased chirality in swimming behaviour. We further demonstrate a macrophage-driven chemotactic strategy involving novel run and tumble behaviour in *Leishmania*, where the parasites change direction, speed, and tumbling behaviour in the presence of human blood-derived macrophages.

## Results and discussion

To characterise motile *L. mexicana*, we harvested PCF and META stages using molecularly verified culture techniques and verified each population pool via morphological distinctions and expression of stage-specific markers, *Histone H4* and *SHERP* (*de Pablos et al., 2019*; *Knuepfer et al., 2001*; *Figure 1A*). We then used holographic microscopy to analyse the three-dimensional motility of suspensions of PCF and META. Subsets of this data are presented as composite renderings of 500 cell tracks in a sample volume $\sim 1.5 \times 1.5 \times 1.2$ mm$^3$ (*Figure 1B, C*, *Video 2*, *Video 3*). In liquid culture, observed

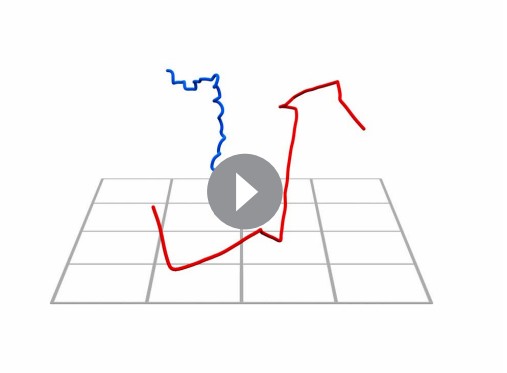

**Video 4.** Computer rendering of two exemplar cell tracks: a procyclic promastigote (blue) and a metacyclic promastigote (red) of *Leishmania mexicana*. The squares on the ground represent 250 µm on a side. The procyclic promastigote track displays the stereotypical meandering helical trajectory, while the metacyclic cell shows longer, straight runs interspersed with tumble events.
https://elifesciences.org/articles/65051#video4

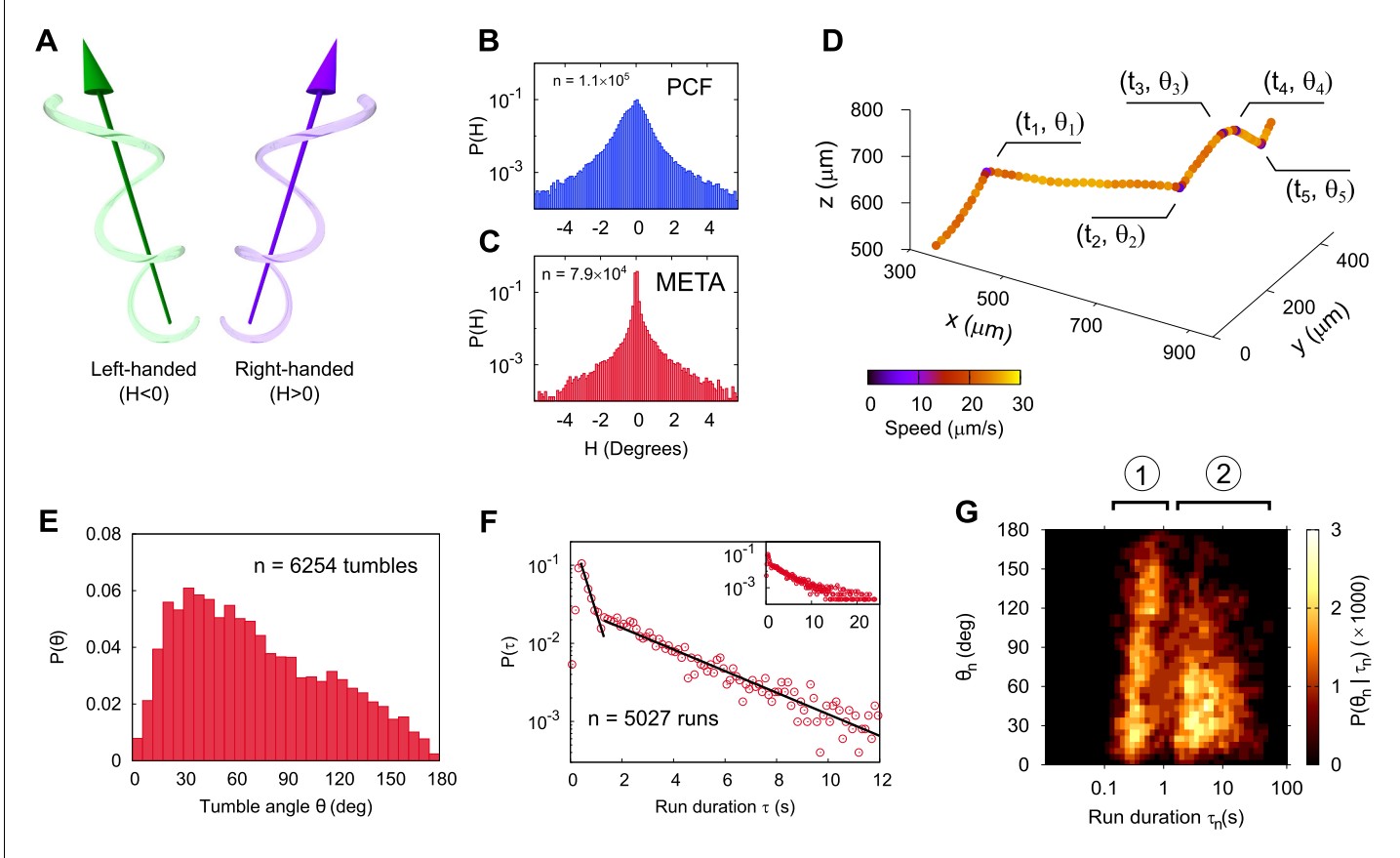

**Figure 2.** Three-dimensional geometry of metacyclic promastigote (META) and procyclic promastigote (PCF) cell tracks in *Leishmania mexicana*. (A) Cartoon depicting left- and right-handed tracks, corresponding to negative and positive helicity, respectively ($H = 0$ is an achiral line). (B) Normalised probability density of instantaneous helicity in a population of PCF ($n = 1.1 \times 10^5$). (C) Normalised probability density of helicity among META ($n = 7.9 \times 10^4$). (D) An example META trajectory in which the colour code shows instantaneous speed. The cell was imaged at 50 Hz, with every 30th point plotted for clarity and five changes in direction ('tumbles'), as indicated. (E) The normalised probability density of tumble angles among META. The mean tumble angle is 72° ± 0.5° (SEM) and the mode is 21°. (F) The distribution of run durations among META, showing two distinct exponential regimes, at short and long times as indicated by the piecewise straight-line fits (black lines). The inset shows the complete range of run durations. (G) Correlation between the duration of a run and the tumble angle following it among META ($n = 5027$ runs). Shorter runs (population 1) are followed by a wider range of angles than longer runs (population 2) indicative of two characteristic run phenotypes.

swimming speeds vary both within and between these two groups. We find the mean swimming speed of PCF cells (*Figure 1D*) to be 18.3 ± 0.2 µm/s (SEM), significantly lower than that of META cells (*Figure 1E*), which have a mean speed of 29.5 ± 0.2 µm/s (SEM). Although the mean swimming speeds are significantly different, the distributions of swimming speeds are broad and overlap: the standard deviations in swimming speed are 8.0 µm/s (PCF) and 10.4 µm/s (META). Such overlap may result from enriched, but not wholly distinct, PCF and META populations being characterised (per gene marker expression) (*Figure 1A,D,E*).

Our results indicate PCF cells swim with a characteristic slow, corkscrew-like motion around a gently curving axis, a similar pattern to that of sea urchin sperm cells (*Jikeli et al., 2015*). In contrast, culture-derived META cells display a distinct swimming phenotype with straight path segments punctuated by sharp turning events, a 'run and tumble' motif reminiscent of enteric bacteria such as *E. coli* (*Figure 1C*). The 'tumble' events were marked by a decrease in the processive speed of the cells, consistent with a reversal of the flagellar beat and similar to pauses in motion observed in trypanosomes (*Heddergott et al., 2012*).

The chirality of swimming motion can be challenging to access experimentally (*Rodríguez et al., 2009*; *Heddergott et al., 2012*; *Wheeler, 2017*; *Schuster et al., 2017*), especially in cells with a complicated morphology. We assess the chirality of the tracks by using three-dimensional tracking

to measure the helicity $H$ (*Wilson et al., 2013*) as illustrated in *Figure 2A* ($H = 0$ implies an achiral line, identical to its mirror image). Importantly, neither PCF (*Figure 2B*) nor META (*Figure 2C*) cell tracks display a systematic chirality on the population level. Individual tracks display a right- or left-handed character, and we do not observe any switches from left- to right-handed chirality (or vice versa) in any of our tracks.

To analyse the characteristic run and tumble nature of META tracks in more detail, we separate the tracks into 'runs' of duration $\tau_n = t_n - t_{n-1}$ punctuated by 'tumbles' through an angle $\theta$ (*Figure 2D*). Tumble angles appear biased towards small angles (*Figure 2E*), so several tumbles would typically be required to completely randomise a cell's swimming direction. In contrast to commonly studied bacterial species that display a heuristically similar swimming pattern, the distribution of *Leishmania* META run durations is not a simple exponential. We observe at least two distinct run behavioural 'regimes', as shown in *Figure 2F*. Runs of $\tau < 1.5$ s are the most common, though the distribution has a long exponential tail. Data on the longest run durations are noisy owing to the finite size of the sample volume, which limits the longest runs that may be captured. The two distinct run regimes are denoted by exponential lines of best fit (non-linear least squares regressions over the domain as indicated) in the main panel of *Figure 2F*, while the inset shows all data down to very low event frequencies at run durations up to 25 s. *Figure 2G* shows the relationship between run duration ($\tau$) and subsequent tumble angle ($\theta$). We note two distinct populations of events, implying two distinct processes: population 1 represents larger reorientations following shorter runs, and population 2 represents slight 'course corrections' during longer runs. This implies that sharper turns are suppressed during longer runs.

In bacterial systems (*Berg, 1993*; *Turner et al., 2016*), an exponential distribution of runs is the hallmark of an underlying Poisson process with a constant probability of tumbling per unit time. Fine-tuning of this tumble probability in response to a biological sensing mechanism enables cells to navigate towards or away from an external stimulus. To determine whether the behaviour of META cells was fixed or could be regulated in conditions that might favour intracellular infection, we repeated our analysis using META cells exposed to J774.2 immortalised mouse macrophages (mMφ) and human monocyte-derived macrophages (hMφ), as well as medium alone (the latter as a negative control), with results shown in *Figure 3*. We compare the number of cells found close to the chemotactic bait ('Tip') to the number found in a region 2 cm from the bait ('Away'), as well as their respective swimming behaviour, in *Figure 3A*. Cells were significantly drawn toward the pipette tip in the presence of primary human blood-derived macrophages, but not to murine macrophages (p < 0.05). Unlike META cells, PCF cells showed a general attraction towards the unrefined agar in the pipette tip even in the absence of a particular stimulus (data shown in *Figure 3—figure supplement 1*).

The number of tumbles exhibited by META cells decreases threefold in the presence of hMφ (*Figure 3B*), independent of the number of tracks observed or track length. A more detailed examination of the geometry of the tracks (quantities recapitulated in *Figure 3C*) yields unaltered distributions of tumble angles (*Figure 3D*) and run durations (*Figure 3E*) between control and hMφ response. This apparently contradictory observation – number of tumbles decreases, but run duration is unaffected – is explained by a sub-population retaining the unstimulated behaviour, while the majority of cells suppress tumbles almost entirely. As described in the Materials and methods section, at least two tumbles per track are required to establish tumble frequency. Therefore, if a cell performs one or zero tumbles, it will not contribute to the run duration statistics. The average displacement per run is enhanced in both 'Tip' and 'Away' cases in the presence of hMφ (*Figure 3F*). Importantly, this indicates the presence of a biologically relevant chemotactic stimulus from hMφ that causes META cells to swim faster and straighter.

Speed distributions for each experimental condition are shown in *Figure 3G–L*. There is no distinction between the 'Tip' and 'Away' speed in the presence of DMEM alone. In contrast, the presence of mMφ somewhat enriches the fraction of cells swimming at low speeds (<10 μm/s), while in the presence of hMφ the average speed is increased by 30%. These distinct phenotypic responses highlight the specificity of the parasite response, and suggest the presence of a soluble, macrophage-derived stimulus to which human-infective META are highly sensitive.

## Summary and conclusions

*Leishmania* cells must survive changing environmental conditions throughout their life cycle, and thus motile stages vary significantly in morphology and behaviour. Pre-adaptation to the next host

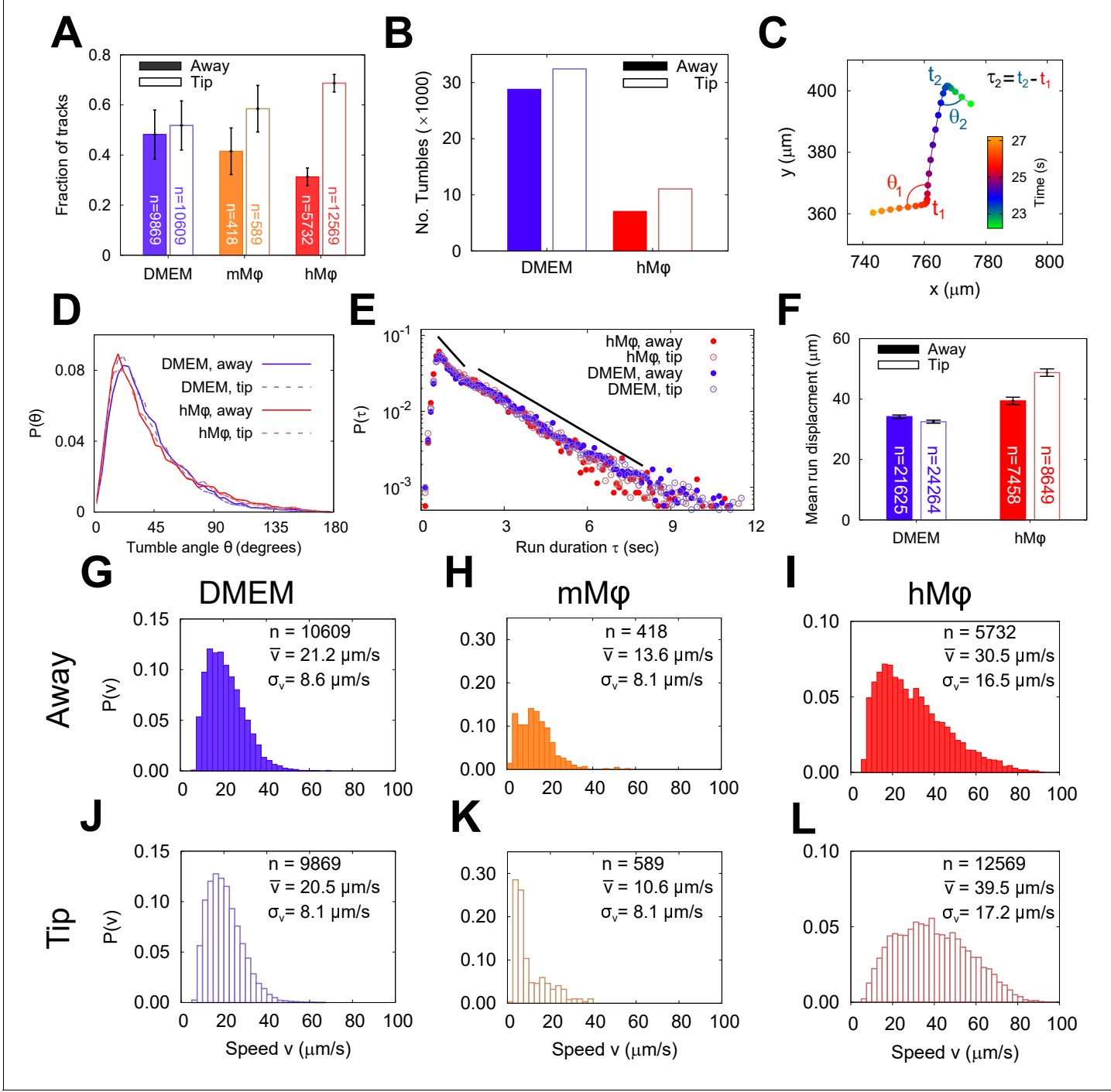

**Figure 3.** The run and tumble behaviour of metacyclic promastigote (META) cells in the presence of an immunological stimulus. (**A**) Fractions of tracks from META cells observed in a bulk suspension of cells 1 cm away from ('Away', solid bars) and immediately adjacent to a loaded pipette tip that had been immersed in the sample medium ('Tip', filled bars) for 30 min. The negative controls, in which tips were filled with DMEM agar, show negligible differences in the number of tracks proximal to the tip, versus the distal bulk. Pipette tips filled with either cultured J774.2 immortalised, mouse stomach-derived macrophages (mMφ) or fresh human blood-derived macrophages (hMφ) showed accumulations of cells around the pipette tips (error bars = 95% CI). (**B**) The number of 'tumble' events for cells in control (DMEM) and test (hMφ) conditions. Exposure to hMφ reduces the number of tumble events considerably, consistent with a more persistent swimming direction. This effect impacts not only the cells that are in the immediate proximity of the human-derived macrophages (hMφ, Tip) but also those in the rest of the sample chamber. As the cells are free to explore the whole sample volume, any 'activation' of cells exposed to a macrophage stimulus may persist even when the cells themselves have left the immediate environment of the pipette tip. (**C**) The x-y projection of a section of a META track, showing tumble times ($t_1, t_2$), reorientation angles ($\theta_1, \theta_2$), and

*Figure 3 continued on next page*

Figure 3 continued

intervening run duration ($\tau_2$). (D) The distribution of tumble angles remains unchanged in the presence of macrophage stimulus, suggesting an unbiased stochastic reorientation process. (E) Distribution of run durations. Contrary to what would be expected in the equivalent situation in chemotaxis in model bacterial systems, there is no increase in run duration in the presence of stimulus (hMφ 'Tip'), compared to controls (hMφ 'Away', DMEM 'Tip', DMEM 'Away'); the distribution of run durations is almost identical in all cases. (F) The mean displacement per run, showing an increase in the run displacement as a consequence of the faster run speed near the macrophage-loaded pipette tip (error bars = error bars=95% CI). (G–L) Normalised instantaneous cell speed distributions for cells distal versus proximal to stimulus. Control samples (DMEM; G,J) show minimal difference, but the presence of mouse- or human-derived macrophages causes a decrease or increase in swimming speed, respectively.

The online version of this article includes the following figure supplement(s) for figure 3:

**Figure supplement 1.** Fluorescein modelling of chemoeffector gradient.
**Figure supplement 2.** Chemotaxis results from procyclic promastigotes (PCF).

environment in parasitic protozoans has been well documented in trypanosomes (*Hill, 2003*; *Rico et al., 2013*). For *Leishmania*, this has been described in terms of changes in metabolism, gene expression (*De Pablos et al., 2016*), and the acquisition of a complement-resistant lipophosphoglycan coat (*McConville et al., 1992*; *Dostálová and Volf, 2012*). We have shown that META have a distinct swimming phenotype and chemotactic response compared to the precursor PCF forms.

In terms of the swimming phenotype, the similarity between *L. mexicana* swimming patterns and those of bacteria is superficial. Bacterial runs and tumbles are a strategy to enable navigation in an environment that randomises their swimming direction through rotational Brownian motion. *Leishmania* cells are at least five to eight times the (linear) size of *E. coli* bacteria, and rotational diffusivity is inversely proportional to particle volume, therefore *Leishmania* will have their orientation randomised by Brownian motion around 100 times more slowly than *E. coli* (*Berg, 1993*). It follows that Brownian motion is negligible when considering *Leishmania* cell orientation, and that the run and tumble behaviour observed here originates from a different imperative, serving a distinct function specific to *Leishmania*. More broadly, we note that foraging strategies among larger animals also follow superficially similar movement patterns (*Viswanathan et al., 1999*).

We find no bias in the chirality of swimming tracks in large populations. This leads us to conclude that helical swimming patterns resulting from small lateral asymmetries or irregularities in cell body shape that are sufficient to cause a rotation through asymmetric fluid drag on the cell body. This natural variation in the shape of cells can be seen in *Video 1*, in which a collection of META cells with beating flagella have been affixed to a glass surface using polylysine. The helical character of the tracks is more obvious in the case of PCF than in META, as seen in *Figure 1F*, *Video 4*. This is borne out in the broader histogram seen in *Figure 2B* compared to that in *Figure 2C*, and is likely a reflection of the larger cell body size relative to flagellar length, compared to META.

The attraction of META to hMφ is perhaps unsurprising given the biological relevance of macrophage infection in the life cycle of this parasite. Both hMφ (the definitive host cells) and neutrophils are drawn to sites of infection during inflammation, while resident dermal macrophages may act as host cells facilitating persistence (*Lee et al., 2020*). Chemical gradients are known to occur and persist in cutaneous infections, as damage to tissue, sandfly saliva and *Leishmania*-derived molecules have been shown to recruit immune cells by this mechanism (*Dey et al., 2018*; *Giraud et al., 2018*; *Alcoforado Diniz et al., 2021*). META cells have also been observed to move within the skin, away from their site of introduction (*Peters et al., 2008*). Under our assay conditions, we did not see a significant attraction to the murine macrophage line J774.2. Previous studies by others have shown that transcriptional responses to pathogens may differ significantly between primary cells and immortalised cells, for example, in the response to *Mycobacterium* (*Andreu et al., 2017*). In addition, the human macrophages used here are relatively quiescent compared to the rapidly cycling J774.2 cells, with likely altered production of metabolites that may affect chemotaxis. Further studies aimed at defining the nature of the chemotactic stimuli involved may help to clarify the observed difference in response.

The control experiments showing an attraction between PCF and DMEM-infused agar are consistent with previous studies on promastigote cultures that had not been subject to META sub-population enrichment (*Oliveira et al., 2000*; *Leslie et al., 2002*; *Barros et al., 2006*). We speculate that this attraction is due to the presence of small carbohydrates associated with the unrefined agar, but note that any subsequent experiments in which PCF might be exposed to hMφ would be biologically

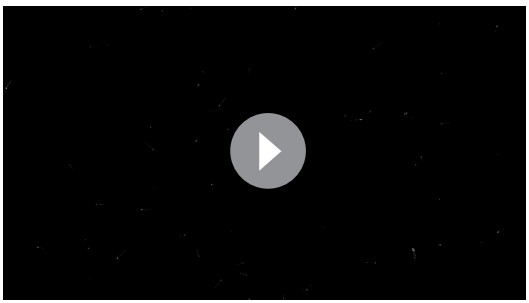

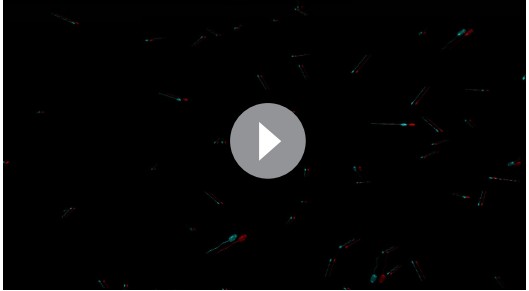

**Video 5.** Artist's impression of a mixed population of swimming metacyclic and procyclic promastigotes of *Leishmania mexicana*. The cells are rendered with the cell bodies in proportion to the flagella, and the cells' swimming trajectories are those obtained from tracking data.

https://elifesciences.org/articles/65051#video5

**Video 6.** Artist's impression of a mixed population of swimming metacyclic and procyclic promastigotes of *Leishmania mexicana*, presented as a three-dimensional (3D) red-cyan anaglyph. The cells are rendered with the cell bodies in proportion to the flagella, and the cells' swimming trajectories are those obtained from tracking data.

https://elifesciences.org/articles/65051#video6

irrelevant. PCF do not naturally encounter and cannot infect hMφ, and attraction towards sugars or other inert matrix components would be difficult to decouple from attraction to cells.

It is of obvious interest to examine how more biologically relevant viscosities might impact these swimming patterns (*Heddergott et al., 2012*). Our in vitro analyses provide a framework to examine *Leishmania* spp. life cycle-dependent motility distinctions in medium, but these parasites encounter dynamic in vivo environments including sandfly bloodmeals, midgut and proboscis and mammalian dermis, bloodstream, and potentially lymph (*Moll et al., 1993*). In addition, *T. brucei* are known to sense each other's presence and this directs movement as evidenced by in vitro social motility studies (*Lopez et al., 2015*; *Imhof et al., 2015*). While the in vivo tsetse fly environment is far more compartmentalised for *T. brucei* life cycle stages than that of the sandfly for *Leishmania*, investigating whether *Leishmania* parasites also display the capacity to sense each other would be an exciting and pertinent tangent.

Our studies show that flagellar motility is modified in cells adapted for infection of the human host. We show that run and tumble behaviour is a phenotypic marker of human-infective META stage parasites, and that this behaviour is suppressed in the presence of a chemotactic stimulus, causing cells to swim faster and straighter towards their target. We speculate that this provides an evolutionary advantage which promotes successful phagocytic uptake of non-replicative META cells, fundamental to human infection and parasite life cycle progression. Through optimised flagellar motility and an as-yet uncharacterised sensing mechanism, *Leishmania* proactively drive their uptake by human immune cells.

## Materials and methods

### *L. mexicana* cell culture

Promastigote parasites of *L. mexicana* (strain M379) were cultured in M199 and Grace's media at 26°C as described previously (*de Pablos et al., 2019*). All cells were maintained within low passage numbers (<5) to ensure biological relevance to differentiation and infectivity was not compromised (for a summary of the *Leishmania* life cycle, see supplementary data to *Figure 1*). PCF were harvested mid-logarithmic phase at concentration 3–6×10⁶ cells/ml. This stock sample was diluted by a factor of 100 into fresh M199 medium to 5×10⁴ cells/ml for holographic tracking. To generate META, PCF culture was passaged into Grace's medium and cultured for 7 days at 26°C (*de Pablos et al., 2019*; *Rogers et al., 2009*). For META populations, cells were centrifuged in a 10% Ficoll gradient (*Späth and Beverley, 2001*) to enrich for META cells as defined by morphology, molecular markers, and macrophage infectivity. Purified META were transferred into fresh M199 medium and diluted to 5×10⁴ cells/ml for holographic tracking.

## Molecular validation of *L. mexicana* promastigote stages via qRTPCR

RNA was isolated from PCF and META populations concurrent to image capture, purified, reverse transcribed, and analysed for *Histone H4* and *SHERP* mRNA levels relative to constitutive *nmt* levels as described previously (*de Pablos et al., 2019*).

## Macrophage cell culture

Immortalised mouse stomach macrophages (J774.2) and human blood-derived macrophages from resident donors were isolated and cultured in vitro for 7 days in Dulbecco's modified eagle medium (DMEM) at 37°C prior to centrifugation. Approximately $1 \times 10^4$ cells were resuspended in 10 µl DMEM:1% agar for each chemotactic assay sample. Macrophage viability was confirmed via alamar blue staining 2 hr after sample preparation.

## Sample chambers

To compare life cycle stages, glass sample chambers were constructed from slides and UV-curing glue, giving a final sample volume measuring approximately $20 \times 15 \times 1.2$ mm$^3$. These were loaded with *L. mexicana* in M199 and sealed with petroleum jelly before observation.

## Chemotaxis assay

The stability of the chemotactic apparatus and resultant gradient was verified via a fluorescein-infused agar gradient establishment test (see supplementary data to *Figure 3*). The fluorescein agar was then replaced in the 10 µl pipette tip by different 'chemotactic bait': DMEM 1% agar alone, DMEM 1% agar with immortalised mouse macrophages or with human blood-derived macrophages. Chemotaxis sample chambers were constructed as above with the addition of the 10 µl pipette tip sealed into the sample chamber with petroleum jelly. The assembly was then placed on a temperature-controlled microscope stage at 34°C in a room set at 34°C to equilibrate for 30 min, allowing transient convection currents to dissipate and a chemical gradient to become established. Cells were imaged immediately adjacent to the pipette tip, but with the tip itself approximately 50 µm outside the field of view to prevent imaging artifacts ('Tip'), and at a distance of approximately 1 cm from the tip ('Away'). Five movies were acquired in each location ('Tip' and 'Away'), alternating between the two locations between individual movie acquisitions.

## Holographic video microscopy

The holographic microscopy setup was similar to that used previously (*Singh et al., 2018*; *Kühn et al., 2018*; *Thornton et al., 2020*), with a few modifications. The samples were imaged on a Nikon Eclipse E600 upright microscope. The illumination source was a single-mode fibre-coupled laser diode with peak emission at 642 nm. The end of the fibre was mounted below the specimen stage using a custom adaptor and delivered a total of 15 mW of optical power to the sample. A Mikrotron MC-1362 monochrome camera was used to acquire videos of 3000 frames, at a frame rate of 50 Hz and with an exposure time of 100 µs. A 10× magnification bright-field lens with numerical aperture of 0.3 was used to acquire data at a video resolution of $1024 \times 1024$ pixels$^2$, corresponding to a field of view measuring $1.44 \times 1.44$ mm$^2$. The raw videos were saved as uncompressed, 8-bit AVI files. Three biological replicates of each condition were prepared, with five technical replicates from each of these.

## Holographic data reconstruction

From each individual video frame, we calculated a stack of 130 images, spaced at 10 µm along the optical axis, sampling the optical field within a volume of $1.44 \times 1.44 \times 1.3$ mm$^3$. These sequences of images were calculated using the Rayleigh-Sommerfeld back-propagation scheme (*Lee and Grier, 2007*). We localised the cells using a method based on the Gouy phase anomaly, as described in more detail elsewhere (*Wilson and Zhang, 2012*; *Farthing et al., 2017*; *Gibson et al., 2021*). This method segments features based on axial optical intensity gradients within a sample, allowing us to extract three-dimensional coordinates for individual cells in each frame. Lateral position uncertainties are approximately 0.5 µm, while the axial performance is slightly worse at approximately 1.5 µm. The latter is limited by the angular resolution of the microscope objective and the scattering properties of the anisotropic cell bodies. A separate software routine was used to compare the putative

cell coordinates extracted in each frame and associate those most likely to constitute cell tracks. The tracks were smoothed using piecewise cubic splines in order to remove noise in the cell coordinates and provide better estimates of cell velocity as described in previous work (*Kühn et al., 2018*). Examining the mean-squared displacement of the cells' smoothed trajectories allowed us to discriminate between swimming and diffusing cells, and to discard the latter. The smoothing process also allowed for linear interpolation of missing data points, up to five consecutive points (equivalent to 0.1 s). Tracks with a duration shorter than 3 s were discarded. To give a more intuitive sense of the swimming trajectories, we have provided an 'artist's impression' rendering of the swimming cells, in *Video 5*, and in a red-cyan anaglyph version in *Video 6*.

## Frequency-domain analysis

To extract the flagellar beat and body roll frequencies, the smoothed trajectories were subtracted from the raw values, and the power spectrum of the residuals was calculated. The power spectra from all cells in the sample were summed to give the data in *Figure 1J*. To illustrate the dominance of the two main frequency components, the track residuals were filtered in the frequency domain using a filter with two Gaussian pass bands with mean (standard deviation) frequencies of 1 Hz (1 Hz) and 20 Hz (2 Hz). The resulting signal contains the body roll and flagellar beat, removing extraneous noise, as shown in *Figure 1H,I*.

## Track chirality analysis

The PCF and META cell tracks were analysed to extract their chirality (handedness). This was done according to a scheme similar to that previously used for determining the chirality of flagellar waveforms (*Wilson et al., 2013*). We first constructed an array of displacement vectors $\mathbf{T}_j$ from a cell's coordinates $\mathbf{r}_j(t)$ by calculating $\mathbf{T}_j = \mathbf{r}_j - \mathbf{r}_{j-1}$, and obtained the local track handedness as the angle formed between vector $\mathbf{T}_j$ and the plane formed by the two previous segments (defined by $\mathbf{T}_{j-2} \wedge \mathbf{T}_{j-2}$), divided by the total contour length $|\mathbf{T}_{j-2}| + |\mathbf{T}_{j-1}| + |\mathbf{T}_j|$. We used this instead of (e.g.) the more conventional Frenet-Serret apparatus because the handedness ($H$) is mathematically well behaved for straight trajectories.

## Run-tumble analysis

Tumbles were identified as large changes in swimming direction coupled with a drop in swimming speed. Quantitatively, we calculated

$$\Xi = 1 - \frac{(v_j + v_{j-1})}{2\langle v \rangle_j} arccos\left( \frac{\mathbf{T}_j \cdot \mathbf{T}_{j-1}}{|\mathbf{T}_j||\mathbf{T}_{j-1}|} \right), \tag{1}$$

where we denote the local speed during segment $\mathbf{T}_j$ as $v_j$ and the average speed of track $j$ as $\langle v \rangle_j$. This quantity shows clear peaks when the cells change direction ('tumbles'), as identified in *Figure 2D*. Run duration $\tau$ was then defined as the time elapsed between two subsequent tumble events. The tumble events take a finite time, during which the cell is essentially stationary. We therefore define the tumble angle as the change in the cell's swimming direction between two time points 1 s apart, with the tumble event halfway in between.

## Acknowledgements

This work was supported by Wellcome Trust grant nos. WT10472 and WT105502MA, EPSRC grant no. EP/N014731/1, and MRC grant no. MR/L00092X/1. We thank Dr Luis M de Pablos for cell culture assistance.

## Additional information

### Funding

| Funder | Grant reference number | Author |
| --- | --- | --- |
| Engineering and Physical | EP/N014731/1 | Laurence G Wilson |

| | | |
|---|---|---|
| Sciences Research Council | | |
| Wellcome Trust | WT104726 | Paul M Kaye |
| Medical Research Council | MR/L00092X/1 | Pegine B Walrad |
| Wellcome Trust | WT105502MA | Rachel C Findlay |
| BBSRC | BB/M011151/1 | Kirstin A Spence |

The funders had no role in study design, data collection and interpretation, or the decision to submit the work for publication.

### Author contributions

Rachel C Findlay, Conceptualization, Data curation, Formal analysis, Validation, Investigation, Methodology, Writing - review and editing; Mohamed Osman, Resources, Investigation; Kirstin A Spence, Visualization; Paul M Kaye, Resources, Formal analysis, Writing - review and editing; Pegine B Walrad, Conceptualization, Resources, Data curation, Formal analysis, Supervision, Funding acquisition, Validation, Investigation, Methodology, Writing - original draft, Project administration, Writing - review and editing; Laurence G Wilson, Conceptualization, Resources, Data curation, Software, Formal analysis, Supervision, Funding acquisition, Investigation, Visualization, Methodology, Writing - original draft, Project administration, Writing - review and editing

### Author ORCIDs

Rachel C Findlay ![ORCID] https://orcid.org/0000-0003-3235-130X
Kirstin A Spence ![ORCID] https://orcid.org/0000-0002-1220-4698
Paul M Kaye ![ORCID] https://orcid.org/0000-0002-8796-4755
Pegine B Walrad ![ORCID] https://orcid.org/0000-0002-2302-0720
Laurence G Wilson ![ORCID] https://orcid.org/0000-0001-6659-6001

### Ethics

Human subjects: Unrelated adults of both genders and diverse racial backgrounds local to Yorkshire, United Kingdom volunteered for tissue sampling for this research. After health screening by medical staff, informed consent was obtained from donors in compliance with ICH GCP, the UK Data Protection Act, and other regulatory requirements, as appropriate and under the University of York Department of Biology Ethics Committee (BEC)-approved study CL 201201 v2.3 to donate blood for a BEC-approved biomedical research under Project License PK 201702.

### Decision letter and Author response

Decision letter https://doi.org/10.7554/eLife.65051.sa1
Author response https://doi.org/10.7554/eLife.65051.sa2

## Additional files

### Supplementary files

• Transparent reporting form

### Data availability

Data underlying the conclusions in this study are are available at the York Research Database (https://doi.org/10.15124/a8eabcd4-66c1-41a3-aa4b-423921b06568).

The following dataset was generated:

| Author(s) | Year | Dataset title | Dataset URL | Database and Identifier |
|---|---|---|---|---|
| Wilson LG | 2021 | Leishmania motility and chemotaxis data | https://doi.org/10.15124/a8eabcd4-66c1-41a3-aa4b-423921b06568 | The York Research Database, 10.15124/a8eabcd4-66c1-41a3-aa4b-423921b06568 |

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
