## [Decision Letter]

**Acceptance summary:**

This study utilizes 3D holographic microscopy to identify differences in the swimming behaviour of different flagellated stages of Leishmania parasites. Infectious metacyclic promastigotes swim faster and have different trajectories compared to non-infectious stages. These stages also alter their trajectories in the presence of macrophages host cells, suggesting that flagellar function and swimming behaviour is influenced by extracellular signals which could potentially increase interactions with target host cells.

**Decision letter after peer review:**

Thank you for submitting your article "High speed, 3D imaging reveals chemotactic behavior specific to human-infective Leishmania parasites" for consideration by *eLife*. Your article has been reviewed by 3 peer reviewers, including Malcolm J McConville as the Reviewing Editor and Reviewer #1, and the evaluation has been overseen by Dominique Soldati-Favre as the Senior Editor. The following individual involved in review of your submission has agreed to reveal their identity: Markus Engstler (Reviewer #2).

Essential revisions:

The three reviews recognized the innovative use of holographic microscopy to investigate Leishmania flagellar function and swimming behavior. However, all reviewers had some concerns about the speculative nature of some of the discussion, particularly statements suggesting that flagellar-dependent chemotaxis might be a unique characteristic of metacyclic promastigotes (rather than a feature of all stages) and the biological relevance of this swimming behavior in more complex tissue environments. The following specific concerns should be addressed:

1. The manuscript should be modified to clearly separate speculative comments on the biological significance of the work from the definitive finding.

2. Please provide additional evidence that only META and not procyclic promastigotes exhibit 'chemotaxis' or remove speculation that chemotaxis may reflect a pre-adaptive response to life in the mammalian host.

3. Please reference additional literature on use of holographic microscopy and flagellar motility in trypanosomatids.

4. Please incorporate additional discussion on chemotaxis in kinetoplastids.

5. Please add further discussion of different META motility response to different macrophage cell types (murine/human).

6. Provide a more critical discussion of the physiological relevance of these findings in vivo (specifically, whether chemotaxis is likely to be relevant when host cells are already being actively recruited to tissue sites in natural sandfly infections, and major host cells that are infected during this period are neutrophils, not macrophages).

*Reviewer #1:*

The authors analyzed the swimming speeds and trajectories of ~500 procyclic (PCF) and purified metacyclic (META) promastigotes using 3D holographic microscopy. This approach allows measurement of speed, trajectories and chirality. Although the META fraction comprised a mixed population of promastigotes, they exhibit a distinctive run and tumble phenotype. The authors then developed an in vitro assay to assess the impact of a potential chemotactic signal, growing host cells, had on META swimming behaviour. They show that META exhibited significant affinity for human primary macrophages (and to a lesser extent J774 murine Mø) compared to medium alone. This was associated with a decrease in tumbling and increase in run duration, allowing directed swimming to the attractant. The use of holographic microscopy to map promastigote swimming trajectories under different conditions is innovative and the finding highlight a novel virulence trait for these protists.

*Reviewer #2:*

Using high-speed holographic methodology, the swimming trajectories of two Leishmania life cycle stages are measured. Significant differences between the life stages become apparent. In addition, the authors show in a chemotaxis experiment that the infectious metacyclics respond chemotactically to the presence of macrophages.

The physics part of the study is flawless, and the holography is very impressive, especially in view of the comparatively simple setup. The analysis and presentation of the data is also flawless.

What is not so clear is the biological interpretation of the data. Chemotactic behavior has been repeatedly postulated for Leishmania, trypanosomes, and other parasites. However, there have been no experiments to date that allow conclusions to be drawn about in vivo relevance. Unfortunately, this does not really change with this study.

It has been shown in trypanosomes that the swimming behavior of different species and life stages are influenced by the mechanical conditions of their microenvironments. Viscosity, obstacles, and hydrodynamics can all play a critical role in determining motility. These factors are ignored in the study. Cell culture medium with the viscosity of water cannot image the situation in the vector or body fluids such as blood or lymph. A chemotactic gradient such as the one generated here by rather simple means cannot arise at all in vivo, simply because everything is in flux and parasites and macrophages move continuously. Moreover, one may wonder why Leishmania should actively move chemotactically toward macrophages when they come into contact with target cells much more rapidly by chance due to self-stirring properties of body fluids. I am not questioning the finding at all. I am merely questioning its biological relevance. Perhaps it would be better to describe this aspect of the paper more cautiously and to discuss it quite openly critically. Otherwise, the result might enter our knowledge as evidence for biologically relevant chemotaxis, and that would be problematic.

The fact that the results are repeatedly mixed with discussion does not make reading the paper any easier either, especially since some of the citations here are suggestive. Important citations are missing and some citations simply do not fit into the context. Here a revision would be appropriate.

L53: Here, for example, doi: 10.1371/journal.pone.0037296 be cited, that to my knowledge was the first holographic study in parasites.

L73: Significant variance in swimming behavior of cells of the same stage is reported. However, the numbers suggest surprisingly little variance.

L79: 'run and tumble': could a stochastic 'beat reversal' of the flagellum be observed?

L82-84: to what extent is Brownian motion a relevant factor at all in the natural environments of Leishmania?

L93 and Figure 2B: How can trajectories be H=O?

Figure 3: The run durations are unchanged. Does this mean that the sole response to the signal is an increase in the frequency of the flagellar beat? This should continue to increase in the gradient. Is this so?

*Reviewer #3:*

The authors describe a clever and powerful assay to show chemotactic behavior in metacyclic Leishmania, which is an important result. The data seem mostly solid, but some results are confusing (perhaps partly an issue with presentation?) and overall conclusions seem like they need to be toned down a little. It is expected that this work will have long-lasting impact on the research community, and the new methods developed will be widely utilized.

• "Pre-Adaptation", e.g. lines 149-150: A major message of the work is to suggest that motility behavior and chemotaxis is a "pre-adaptation". However, I don't agree that the current studies show that "…flagellar motility is a.…preadaptation to infection of human hosts." What are the data to support this? The authors do a very good job of defining motility features of PCF and META forms, including quantitative analysis of motility features in 3D. They find that motility differs in PCF vs META forms. They also demonstrate chemotaxis in META forms. But, I don't see how these combined results demonstrate a "pre-adaptation" to infection of human hosts. As such, the "pre-adaptation" statement should be moved to speculation. Notably, I did not see tests for chemotaxis in PCF. Thus, it is even not formally demonstrated whether or not chemotaxis itself is an "adaptation" specific to META forma, or rather (and quite likely) is a fundamental property of all life cycle stages.

– To test if chemotaxis is an 'adaptation', the authors would need to provide an analysis of PCFs. To be an adaptation, one would expect to find either that PCFs do not exhibit chemotaxis, or that they do not chemotax toward macrophages in the assay used. Without this, the authors cannot say whether chemotaxis is a stage-specific behavior, much less a "pre"-adaptation.

– Note, I think the work would not be negatively affected if the whole concept of "adaptation" were omitted and the work was framed around the very important results of developing a new and powerful approach to investigate Leishmania motility in 3D; quantitative definition of motility parameters; demonstration of chemotaxis in META forms.

• Chemotaxis: The work would benefit from some commentary on chemotaxis in kinetoplastids. A 'suggestion' for a potential advantage provided by chemotaxis (lines153-155) is not unwarranted, but that should be kept to speculation at this point, and implication that this is an 'adaptation' is not supported by the current data. With report of chemotaxis being a major message, the paper would benefit from a brief discussion on what's been demonstrated regarding chemotaxis in trypanosomatids, as this is an important, yet under-represented area of research on these organisms. Without this, the novelty and significance of the author's rigorous, novel and very interesting work are not brought out.

• Lines 125 – 129: How is it that tumble frequency decreases, but run duration is unaffected? I would think that less frequent tumbles would lead to longer runs? This warrants more comment.

• Figure 3 and Lines 135-139: How does one reconcile the finding that murine macrophages and human macrophages both induce taxis toward the pipet tip (3A), but there is opposite impact on speed profiles, with murine macrophages causing slower speeds, and human macrophages causing faster speeds (3H,K vs 3I,L)? Perhaps analysis done for human macrophages must also be done for murine macrophages. Some more commentary, and analysis needs to be provided on this point.

• Regarding replicates: While the number of cells tracked are clearly indicated, I did not see a description of how many different chambers were imaged for each condition, or how many different fields per chamber.

---

## [Author Response]

Essential revisions:The three reviews recognized the innovative use of holographic microscopy to investigate Leishmania flagellar function and swimming behavior. However, all reviewers had some concerns about the speculative nature of some of the discussion, particularly statements suggesting that flagellar-dependent chemotaxis might be a unique characteristic of metacyclic promastigotes (rather than a feature of all stages) and the biological relevance of this swimming behavior in more complex tissue environments. The following specific concerns should be addressed:1. The manuscript should be modified to clearly separate speculative comments on the biological significance of the work from the definitive finding.

We have separated the Results and the Discussion sections to clarify outcomes from contextual interpretation.

2. Please provide additional evidence that only META and not procyclic promastigotes exhibit 'chemotaxis' or remove speculation that chemotaxis may reflect a pre-adaptive response to life in the mammalian host.

Apologies for not including these results in the original submission; we have also moderated the claim about pre-adaptation (in the abstract, and around lines 188-90 in the summary/conclusions). The PCF promastigote results have now been included in supplementary data (Figure 3—figure supplement 2). PCF cells show a non-specific attraction towards agar-filled pipette tips (which we speculate is due to the sugars released from the unrefined agar matrix). These findings are consistent with previous studies of chemotaxis in promastigotes, in experiments that did not enrich the metacyclic subpopulation.

3. Please reference additional literature on use of holographic microscopy and flagellar motility in trypanosomatids.

We have enhanced our discussion of holographic microscopy in the introduction (lines 82-99), and its application to trypanosomatids. We have added 3 additional references to the manuscript (Rodriguez et al. 2009, Romero et al. 2012, Weiße et al. 2012).

4. Please incorporate additional discussion on chemotaxis in kinetoplastids.

We have added an extra paragraph in the introduction providing more context and references about chemotaxis in kinetoplastids (lines 53-81).

5. Please add further discussion of different META motility response to different macrophage cell types (murine/human).

We have expanded the Discussion, and thank the reviewer for raising this issue. The response to murine macrophages is not significant. We have contextualised this result in the manuscript (lines 159-164, 215-229).

6. Provide a more critical discussion of the physiological relevance of these findings in vivo (specifically, whether chemotaxis is likely to be relevant when host cells are already being actively recruited to tissue sites in natural sandfly infections, and major host cells that are infected during this period are neutrophils, not macrophages).

We thank the reviewers for pointing this out, and have expanded the discussion to provide a fuller explanation of these aspects. Whilst we agree that studies demonstrate rapid neutrophil influx at the site of sand fly bite, other critical populations for *Leishmania* persistence e.g. resident dermal macrophages (Lee et al. Sci Immunol. 2020 Apr 10;5(46)) are generally believed to be relatively immobile.

Reviewer #2:Using high-speed holographic methodology, the swimming trajectories of two Leishmania life cycle stages are measured. Significant differences between the life stages become apparent. In addition, the authors show in a chemotaxis experiment that the infectious metacyclics respond chemotactically to the presence of macrophages.The physics part of the study is flawless, and the holography is very impressive, especially in view of the comparatively simple setup. The analysis and presentation of the data is also flawless.What is not so clear is the biological interpretation of the data. Chemotactic behavior has been repeatedly postulated for Leishmania, trypanosomes, and other parasites. However, there have been no experiments to date that allow conclusions to be drawn about in vivo relevance. Unfortunately, this does not really change with this study.It has been shown in trypanosomes that the swimming behavior of different species and life stages are influenced by the mechanical conditions of their microenvironments. Viscosity, obstacles, and hydrodynamics can all play a critical role in determining motility. These factors are ignored in the study. Cell culture medium with the viscosity of water cannot image the situation in the vector or body fluids such as blood or lymph. A chemotactic gradient such as the one generated here by rather simple means cannot arise at all in vivo, simply because everything is in flux and parasites and macrophages move continuously. Moreover, one may wonder why Leishmania should actively move chemotactically toward macrophages when they come into contact with target cells much more rapidly by chance due to self-stirring properties of body fluids. I am not questioning the finding at all. I am merely questioning its biological relevance. Perhaps it would be better to describe this aspect of the paper more cautiously and to discuss it quite openly critically. Otherwise, the result might enter our knowledge as evidence for biologically relevant chemotaxis, and that would be problematic.

We thank the reviewer for their perspective and agree that providing formal evidence for chemotaxis in vivo is complicated. The reviewer is right that mechanical stimulus, viscosity, elasticity etc. are present in body tissues, and that they will affect the motion of the flagellum, and that there is evidence that physical obstructions interrupt the flagellar beat (though ‘stirring’ does not play a role in Leishmania’s motion through tissue). At any rate, we contend that an in vitro study such as ours decouples the mechanical heterogeneity of the in vivo environment from the parasite’s cellular response. If a chemotactic response is present in the parasite, then it will be most sensitively and uniquely tested in an isotropic environment such as a bulk Newtonian fluid – indeed, this is what we find. Chemical gradients are known to occur and persist in cutaneous infections, as damage to tissue, sand fly saliva and *Leishmania*-derived molecules have been shown to recruit immune cells by this mechanism – we have added references and words to this effect on lines 211-214.

The fact that the results are repeatedly mixed with discussion does not make reading the paper any easier either, especially since some of the citations here are suggestive. Important citations are missing and some citations simply do not fit into the context. Here a revision would be appropriate.L53: Here, for example, doi: 10.1371/journal.pone.0037296 be cited, that to my knowledge was the first holographic study in parasites.

We thank the reviewer for this suggestion and have included the suggested reference at this point in the text.

L73: Significant variance in swimming behavior of cells of the same stage is reported. However, the numbers suggest surprisingly little variance.

We have refined this statement in light of the reviewer’s remark. Importantly, the lifecycle stage populations examined are enriched but as the molecular markers suggest in Figure 1A, not purely distinct. Despite this, the mean speed of META population is still around 50% larger than that of PCF. The uncertainties quoted (on the order of 0.2 um/s) are errors on the mean. We have clarified these metrics in the text, and included the standard deviations in swimming speed as a way of characterising the breadth of the distributions. By these standards, the variation is significant.

L79: 'run and tumble': could a stochastic 'beat reversal' of the flagellum be observed?

Yes, we believe that this is the case. The processive speed of the cells decreases noticeably around the time of the tumbles, consistent with a flagellar beat reversal. The current assays were designed to maximise the volume imaged, rather than spatial resolution however, so we do not have details of the beat reversal dynamics. We have amended the text to include this.

L82-84: to what extent is Brownian motion a relevant factor at all in the natural environments of Leishmania?

In terms of re-orienting the cells, Brownian motion is likely irrelevant for Leishmania. It’s therefore surprising that their swimming phenotype is superficially similar to that of bacteria, which certainly *are* affected by Brownian motion. We’ve strengthened this statement (rephrased the statement containing the word ‘advisedly’) accordingly to avoid confusion. Brownian motion is relevant for the distribution and diffusion of chemical messengers in the cells’ environment (and from our pipette tip), but given the context we don’t think that this was the reviewer’s point.

L93 and Figure 2B: How can trajectories be H=O?

A helicity of zero implies an achiral curve, i.e. one that is neither left- or right-handed (which would have negative or positive sign respectively). We have modified the text in the two locations indicated to clarify this.

Figure 3: The run durations are unchanged. Does this mean that the sole response to the signal is an increase in the frequency of the flagellar beat? This should continue to increase in the gradient. Is this so?

This isn’t quite the case; we’ve expanded the discussion in response to this comment and the one from Reviewer #3. Essentially, most cells suppress tumbles, while a few cells retain the unstimulated response, and these ‘unstimulated’ cells dominate the run duration statistics. Such ‘unstimulated’ cells may be deficient in responsive factors given the genetic variability noted in *Leishmania* cultures (Rogers et al., 2014)

Reviewer #3:The authors describe a clever and powerful assay to show chemotactic behavior in metacyclic Leishmania, which is an important result. The data seem mostly solid, but some results are confusing (perhaps partly an issue with presentation?) and overall conclusions seem like they need to be toned down a little. It is expected that this work will have long-lasting impact on the research community, and the new methods developed will be widely utilized.• "Pre-Adaptation", e.g. lines 149-150: A major message of the work is to suggest that motility behavior and chemotaxis is a "pre-adaptation". However, I don't agree that the current studies show that "…flagellar motility is a.…preadaptation to infection of human hosts." What are the data to support this? The authors do a very good job of defining motility features of PCF and META forms, including quantitative analysis of motility features in 3D. They find that motility differs in PCF vs META forms. They also demonstrate chemotaxis in META forms. But, I don't see how these combined results demonstrate a "pre-adaptation" to infection of human hosts. As such, the "pre-adaptation" statement should be moved to speculation. Notably, I did not see tests for chemotaxis in PCF. Thus, it is even not formally demonstrated whether or not chemotaxis itself is an "adaptation" specific to META forma, or rather (and quite likely) is a fundamental property of all life cycle stages.– To test if chemotaxis is an 'adaptation', the authors would need to provide an analysis of PCFs. To be an adaptation, one would expect to find either that PCFs do not exhibit chemotaxis, or that they do not chemotax toward macrophages in the assay used. Without this, the authors cannot say whether chemotaxis is a stage-specific behavior, much less a "pre"-adaptation.

We have moderated the language around claims of ‘pre-adaptation’ (please see next point for locations), and provided additional results from chemotaxis assays in PCF. Consistent with previous studies (e.g. Oliveira et al., Exp. Parasitol. (2000), Leslie et al., Exp. Parasitol. (2002), Barros et al., Exp. Parasitol. (2006)), we find a different chemo/osmotactic response in which PCF cells are drawn towards the agar in the pipette tip even in the absence of an embedded stimulant such as macrophages. We speculate that this result is due to the presence of small carbohydrate molecules from the unrefined agar – and note that the response is distinct to META, which show no such attraction. However, as suggested, this has been made more speculative in the revised discussion.

– Note, I think the work would not be negatively affected if the whole concept of "adaptation" were omitted and the work was framed around the very important results of developing a new and powerful approach to investigate Leishmania motility in 3D; quantitative definition of motility parameters; demonstration of chemotaxis in META forms.

We thank the reviewer for their suggestion (and their positive words), and have modified the language around claims of pre-adaptation. We have rephrased the claims in the abstract, and around lines 188-90 in the summary/conclusions.

• Chemotaxis: The work would benefit from some commentary on chemotaxis in kinetoplastids. A 'suggestion' for a potential advantage provided by chemotaxis (lines153-155) is not unwarranted, but that should be kept to speculation at this point, and implication that this is an 'adaptation' is not supported by the current data. With report of chemotaxis being a major message, the paper would benefit from a brief discussion on what's been demonstrated regarding chemotaxis in trypanosomatids, as this is an important, yet under-represented area of research on these organisms. Without this, the novelty and significance of the author's rigorous, novel and very interesting work are not brought out.

We thank the reviewer for this suggestion, and have added another paragraph to the introduction (lines 53-81), giving additional context to our results by providing an overview of more experiments in the field. We have also changed the word ‘suggest’ to ‘speculate’ in the summary and conclusions (line 243).

• Lines 125 – 129: How is it that tumble frequency decreases, but run duration is unaffacted? I would think that less frequent tumbles would lead to longer runs? This warrants more comment.

We thank the reviewer for pointing out the apparent confusion here. This stems from the fact that (as stated in the subsequent sentence) in the majority of the population, the tumble rate is significantly suppressed, to either one or zero tumbles per track. We require at least two tumbles per track to measure run duration, so the small fraction of the population unaffected by the stimulus contributes the bulk of the measurable runs. We have clarified this section of the text to clarify how we measure run duration.

• Figure 3 and Lines 135-139: How does one reconcile the finding that murine macrophages and human macrophages both induce taxis toward the pipet tip (3A), but there is opposite impact on speed profiles, with murine macrophages causing slower speeds, and human macrophages causing faster speeds (3H,K vs 3I,L)? Perhaps analysis done for human macrophages must also be done for murine macrophages. Some more commentary, and analysis needs to be provided on this point.

We thank the reviewer for this suggestion, and in the light of their comments, we have revised our description of the murine data, highlighting that the results are not statistically significant. To further emphasise this point to the reader, we have recast the error bars in figure 3a in terms of 95% confidence intervals rather than using the standard error on the mean, as in the previous version. Although one may be calculated directly from the other without any further assumptions, the 95% CI representation might be more familiar to the readership. In this light, the fairly modest decrease in average swimming speed (also seen in absolute terms in the DMEM case) reinforces the revised conclusion that the null hypothesis (META are not stimulated by mm\phi) cannot be rejected.

• Regarding replicates: While the number of cells tracked are clearly indicated, I did not see a description of how many different chambers were imaged for each condition, or how many different fields per chamber.

This has been amended in the Methods section, subheading “Chemotaxis Assay”